# Towards an uncertainty evaluation model for marine track considering local sampling cloud and sample information

Cheng Fang[1,2,4], Wei Zhou[3,4], Xinguo Liu[1], Hongxiang Ren[5], Jianjun Wu[6]*

1 College of Computer Science and Technology, Zhejiang University, Hangzhou, China, 2 Zhejiang Scientific Research Institute of Transport, Hangzhou, China, 3 College of Information Engineering, Zhejiang University of Technology, Hangzhou, China, 4 College of Marine, Zhejiang Institute of Communications, Hangzhou, China, 5 Avigation College, Dalian Maritime University, Dalian, China, 6 Merchant Marine College, Shanghai Maritime University, Shanghai, China

* jjwu@shmtu.edu.cn

**Data Availability Statement:** All relevant data are within the paper and its Supporting information files.

## Abstract

To evaluate the practical ability of crews during the navigation of an inward-port single ship, a track evaluation model was developed on a planar forward normal cloud chart under sample information based on the forward normal and the backward normal cloud generator. Since the track sampling cloud may be too divergent, a track belt division method based on the contributions of normal cloud drops was proposed. Combining the track evaluation model with the track belt division method, a comprehensive track evaluation scheme of the local sampling cloud based on sampling information was established. The results of an example of M.V. DAQING 257 unloaded into Dalian Port demonstrated the effectiveness of the model and showed its consistency with expert evaluation results based on subjective information. The proposed uncertainty evaluation model provides a new approach for intelligent evaluation under sample information.

## 1 Introduction

In the voyage training or examination of inward-port single ships using a ship-handling simulator, a certain amount of objective information is left in the electronic chart of the ship-handling simulator [1, 2]. The objective information related to the ship involves the ship's name, call sign, captain, ship width, draft, ship speed, heading, ship longitude and latitude position, wind current magnitude, wind current direction, loading condition, and rudder angle size [3]. Subjective evaluation results of the assessor are stored in the database after evaluation. The main purpose of intelligent evaluation of inward-port single ships is to make full use of all the data based on sample information.

Scholars studying the evaluation system of ship-handling simulators are mainly centralized in the Dalian Maritime University and Jimei University. In view of evaluation methods, most studies adopted the combinations of expert experience and membership function [4–11], the fuzzy comprehensive evaluation method [12, 13], the virtual reality combination method [11–

**Funding:** This research was supported by the Zhejiang Provincial Natural Science Foundation of China under Grant No. LGG18E090001, the Zhejiang Provincial Postdoctoral Research Project under Grant 225846, and the Open Re-search Project of the State Key Laboratory of Industrial Control Technology, Zhejiang University, China (No. ICT2021B16).

**Competing interests:** The authors have declared that no competing interests exist.

14], and other intelligent evaluation AHP methods [15–19]. In 2011, Chen [4] developed a set of intelligent examination systems based on marine radar/automatic radar plotting aids (ARPA) using a combination of expert experience and membership function. In 2012, Chen [5] developed a radar plotting evaluation model based on the encounter situation, achieving a favorable effect in a radar simulator. In 2015, Liu realized an operational automatic evaluation of marine VHF communication equipment using the analytic hierarchy process (AHP) [19–27] and distribution evaluation method [15]. Jiang [16] studied the simulated operational evaluation of a marine narrow band direct printing telegraph (NBDP). In 2016, Jiang [13] proposed a combination method of expert evaluation and fuzzy comprehensive evaluation and developed an automatic evaluation system for mooring control. Li [18] established an evaluation model and integrated it into a radar plotting training and automatic evaluation system by setting reasonable threshold values, weights, and membership functions according to expert experience.

Based on the above literature, it can be concluded that expert experience, membership function, and fuzzy comprehensive methods are the most commonly adopted methods in evaluation system studies of ship-handling simulators. The evaluation algorithms are relatively single, with only the fuzziness of subjective information such as expert experience considered, but not the randomness, imperfection, and incompleteness in uncertain information. In other words, there are still few studies on evaluation methods that use objective sample information.

The cloud model theory considers the overall fuzziness, randomness, and relevance between the two [28–31], of which the backward normal cloud generator algorithm is one of the most important algorithms [32–36]. The backward cloud learning algorithm makes intelligent evaluation based on big data possible by inferring three key parameters of the cloud model from limited objective track sample information and generating large sample data by using a forward cloud generator.

In this study, an intelligent evaluation model of a normal track cloud under sample information was developed based on the backward normal cloud algorithm with uncertainty and the forward normal cloud generator. To control the divergence of the track sampling cloud, a track belt division method based on the cloud drop contribution of a normal cloud was constructed. Then, an intelligent comprehensive evaluation model of a local sampling track cloud based on sample information was proposed, and its accuracy and efficiency were verified through experimental grading of the track to be evaluated.

## 2 Model establishment

### 2.1 Track evaluation model of inward-port single ship based on local sampling cloud chart

For preliminary knowledge, the maximum boundary curve, minimum boundary curve, and expectation curve of the one-dimensional forward normal cloud model *cloud*(*Ex,En,He,N*) = *cloud*(0.5,0.15,0.03,3000) are shown in Fig 1.

It can be inferred that the expectation curve is a desirable curve in general. Given an unevaluated and objective track sample data in the database, a track evaluation model of an inward-port single ship of a local sample cloud chart is proposed in this section as follows.

Input: unevaluated track $H_d$ and N objective sample tracks stored in the database.

Output: scores of unevaluated track $H_{dscore}$.

**(1)** Track data sampling: Analyze actual navigation data and eliminate track lines whose deviations from the navigation position to the planned position are particularly large (considered as unreasonable track lines). This is followed by sampling M points from the starting point to the end point at equal time intervals. The point β of the track line α can then be

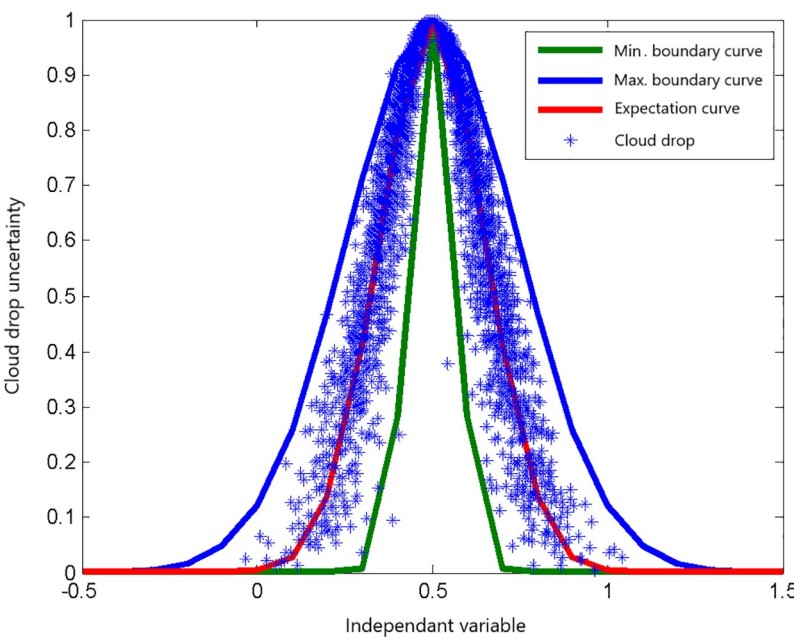

**Fig 1. The schematic diagram of boundary curve and expectation curve of the model *cloud*(0.5,0.15,0.03,3000).**

represented as

$$point_{\alpha\beta}, \alpha = 1, 2, \cdots, \beta = 1, 2, \cdots, M \tag{1}$$

**(2)** Generate a planar forward normal cloud chart of the track sampling points. For all points $\beta$ of $N$ objective sample tracks, define Point $\beta$ as $point_\alpha(\varphi_\beta, \lambda_\beta)$ to form the following set:

$$S_\beta = \{point_1(\varphi_\beta, \lambda_\beta), point_2(\varphi_\beta, \lambda_\beta), \cdots, point_\alpha(\varphi_\beta, \lambda_\beta)\} \tag{2}$$

For $S_\beta$, we have

$$E\hat{x}_\beta = \overline{\varphi}_\beta = \frac{1}{N}\sum_{\alpha=1}^{N}\varphi_{\alpha\beta}, E\hat{y}_\beta = \overline{\lambda}_\beta = \frac{1}{N}\sum_{\alpha=1}^{N}\lambda_{\alpha\beta} \tag{3}$$

$$En\hat{x}_\beta = \sqrt{\frac{\pi}{2}}\frac{1}{N}\sum_{\alpha=1}^{N}|\varphi_{\alpha\beta} - \overline{\varphi}_\beta|, En\hat{y}_\beta = \sqrt{\frac{\pi}{2}}\frac{1}{N}\sum_{\alpha=1}^{N}|\lambda_{\alpha\beta} - \overline{\lambda}_\beta| \tag{4}$$

$$(S\hat{x}_\beta)^2 = \frac{1}{N-1}\sum_{\alpha=1}^{N-1}(\varphi_{\alpha\beta} - \overline{\varphi}_\beta)^2, (S\hat{y}_\beta)^2 = \frac{1}{N-1}\sum_{\alpha=1}^{N-1}(\lambda_{\alpha\beta} - \overline{\lambda}_\beta)^2 \tag{5}$$

$$He\hat{x}_\beta = \sqrt{(S\hat{x}_\beta)^2 - (En\hat{x}_\beta)^2}, He\hat{y}_\beta = \sqrt{(S\hat{y}_\beta)^2 - (En\hat{y}_\beta)^2} \tag{6}$$

Thus, the forward normal planar cloud model of Point $\beta$ can be represented as

$$cloud_\beta = (E\hat{x}_\beta, E\hat{y}_\beta, En\hat{x}_\beta, En\hat{y}_\beta, He\hat{x}_\beta, He\hat{y}_\beta) \tag{7}$$

A planar normal random number $(En\hat{x}'_\beta, En\hat{y}'_\beta)$ can be generated with the expected value as $(En\hat{x}_\beta, En\hat{y}_\beta)$ and the variance as $(He\hat{x}_\beta, He\hat{y}_\beta)$ based on this model. Subsequently, a planar

normal random number $drop(\varphi_{\beta i}, \lambda_{\beta,i})$ can be generated with the excepted value as $(E\hat{x}_\beta, E\hat{y}_\beta)$ and the variance as $(En\hat{x}'_\beta, En\hat{y}'_\beta)$. The cloud drop certainty $\mu_{\beta i}$ can be calculated as follows:

$$\mu_{\beta i} = \exp\left(-\left(\frac{(\varphi_{\beta i} - E\hat{x}_\beta)^2}{2(En\hat{x}'_\beta)^2} + \frac{(\lambda_{\beta i} - E\hat{y}_\beta)^2}{2(En\hat{y}'_\beta)^2}\right)\right) \tag{8}$$

Assuming that each sampling track point needs to output $K$ cloud drops, the cloud chart of $K$ track points of the sampling point $\beta$ can be generated by the forward normal cloud generator above. Similarly, $M$ planar forward normal cloud charts of all sampling points in the track line can be accomplished.

**(3)** Determine the reference surface function of the planar forward normal cloud of an unevaluated track. First, we perform equal time interval sampling for $M$ points in the unevaluated line $H_d$, and denote the sampling point $\beta$ as $point_D(\varphi_{D\beta}, \lambda_{D\beta})$. Second, we calculate the reference surface function of the planar forward normal cloud, including the maximum boundary surface function $f(\varphi_{D\beta}, \lambda_{D\beta}, Z\max_{D\beta})$, expected surface function $f(\varphi_{D\beta}, \lambda_{D\beta}, Z_{D\beta})$, and modified expected surface function $f(\varphi_{RD\beta}, \lambda_{RD\beta}, Z_{RD\beta})$. The corresponding $Z\max_{D\beta}$, $Z_{D\beta}$ and $Z_{RD\beta}$ are calculated as follows:

$$Ex_\beta = \overline{\varphi_\beta} = \frac{1}{N}\sum_{\alpha=1}^{N}\varphi_{\alpha\beta}, Ex_\beta = \overline{\lambda_\beta} = \frac{1}{N}\sum_{\alpha=1}^{N}\lambda_{\alpha\beta} \tag{9}$$

$$En\hat{x}_\beta = \sqrt{\frac{\pi}{2}}\frac{1}{N}\sum_{\alpha=1}^{N}|\varphi_{\alpha\beta} - \overline{\varphi_\beta}|, En\hat{y}_\beta = \sqrt{\frac{\pi}{2}}\frac{1}{N}\sum_{\alpha=1}^{N}|\lambda_{\alpha\beta} - \overline{\lambda_\beta}| \tag{10}$$

$$(S\hat{x}_\beta)^2 = \frac{1}{N-1}\sum_{\alpha=1}^{N-1}(\varphi_{\alpha\beta} - \overline{\varphi_\beta})^2, (S\hat{y}_\beta)^2 = \frac{1}{N-1}\sum_{\alpha=1}^{N-1}(\lambda_{\alpha\beta} - \overline{\lambda_\beta})^2 \tag{11}$$

$$He\hat{x}_\beta = \sqrt{(S\hat{x}_\beta)^2 - (En\hat{x}_\beta)^2}, He\hat{y}_\beta = \sqrt{(S\hat{y}_\beta)^2 - (En\hat{y}_\beta)^2} \tag{12}$$

$$Z_{D\beta} = \exp\left(-\left(\frac{(\varphi_{D\beta} - Ex_\beta)^2}{2(Enx_\beta)^2} + \frac{(\lambda_{D\beta} - Ex_\beta)^2}{2(Eny_\beta)^2}\right)\right) \tag{13}$$

$$Z\max_{D\beta} = \exp\left(-\left(\frac{(\varphi_{D\beta} - Ex_\beta)^2}{2(Enx_\beta + 3He\hat{x}_\beta)^2} + \frac{(\lambda_{D\beta} - Ex_\beta)^2}{2(Eny_\beta + 3He\hat{y}_\beta)^2}\right)\right) \tag{14}$$

$$Z_{RD\beta} = \exp\left(-\left(\frac{(\varphi_{RD\beta} - Ex_\beta)^2}{2(Enx_\beta + He\hat{x}_\beta)^2} + \frac{(\lambda_{RD\beta} - Ex_\beta)^2}{2(Eny_\beta + He\hat{y}_\beta)^2}\right)\right) \tag{15}$$

The reference standard of the excepted surface and the maximum boundary surface can be set based on different needs. Thus, the evaluation value of the sampling point $\beta$ is $Z_{D\beta} \times 100$, $Z\max_{D\beta} \times 100$ or $Z_{RD\beta} \times 100$.

**(4)** Compute the evaluation result of the unevaluated track $H_{dscore}$ comprehensively.

$$H_{dscore} = \frac{1}{M} \sum \beta = 1^M Z_{D\beta} \times 100$$

$$orH_{dscore} = \frac{1}{M} \sum \beta = 1^M Zmax_{D\beta} \times 100 \tag{16}$$

$$orH_{dscore} = \frac{1}{M} \sum \beta = 1^M Z_{RD\beta} \times 100$$

## 2.2 Track belts division based on cloud drops contribution of normal cloud

In ship navigation, its positions and tracks are obvious results of ship handling [37]. Consequently, ship navigation is performed by comparing the position of the ship with that of the same sampling point in history sampling data. Additionally, dividing the corresponding track belt precautionary area is another favorable approach for navigation evaluation and sailor reminding.

From the importance division of normal cloud drops in Fig 2, the contributions of normal cloud drops to the concept are as follows: Cloud drops $[Ex - 0.67En, Ex + 0.67En]$ contribute 50% to concept, representing 22.33% share of all quantitative values, referred to as key elements. Cloud drops $[Ex - En, Ex + En]$ contribute 68.26% to concept, representing a 33.33% share of all quantitative values, referred to as basic elements. Cloud drops $[Ex - 2En, Ex + 2En]$ contribute 95.44% to concept, representing 66.66% of all quantitative values. Cloud drops $[Ex - 3En, Ex + 3En]$ contribute 99.74% to concept, representing a 99.99% share of all quantitative values. The elements in interval $[Ex - 2En, Ex - En]$ and $[Ex + En, Ex + 2En]$ are referred to as

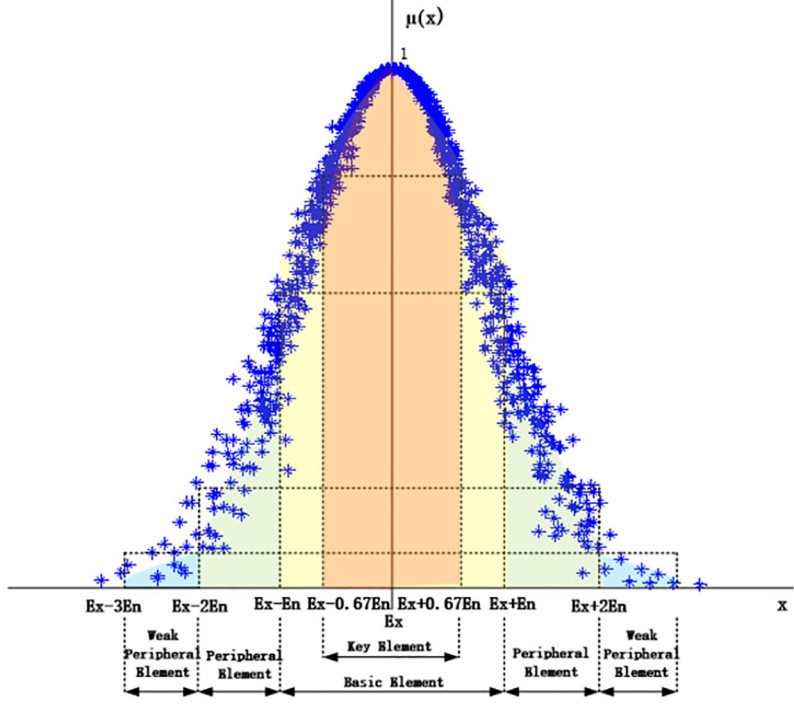

**Fig 2. The importance division of normal cloud drops.**

the peripheral elements. The elements in the interval [*Ex* − 3*En*, *Ex* − 2*En*] and [*Ex* + 2*En*, *Ex* + 3*En*] are referred to as weak peripheral elements.

According to the above analysis, the track belts are divided based on the cloud drop contribution of the normal cloud, and the contributions of the key element belt, basic element belt, peripheral element belt, and weak peripheral element belt are 50%, 68.26%, 95.44%, and 99.74%, respectively. Because there are few differences between the contributions of the peripheral and weak peripheral element belts, the division of the key element belt, the basic element belt, and the peripheral belt are mainly considered.

A comparative analysis of the belt distribution and channel width can be used to further improve the inward-port single-ship track evaluation model based on local sampling cloud chart. However, there are some differences among the evaluations of different navigation processes, such as navigation in straight channels or in curved channels. In the former cases, it is generally impermissible to navigate out of the channel except under emergency or other specific causes, while in the latter cases, it is normal to have divergent navigation positions due to the influences of cycle performance, vehicle, and rudder performance of the ship. When tracks are out of the channel, the score-reducing scheme, vote-overrule system, and disqualification should be adopted according to specific situations and the assessor's requirement, which is integrated into the evaluation model of an inward-port single ship based on a local sampling cloud chart.

## 2.3 Comprehensive track evaluation model based on sample information

The evaluation process of the inward-port single-ship trajectory evaluation based on the sample information is shown in Fig 3. It consists of the following steps.

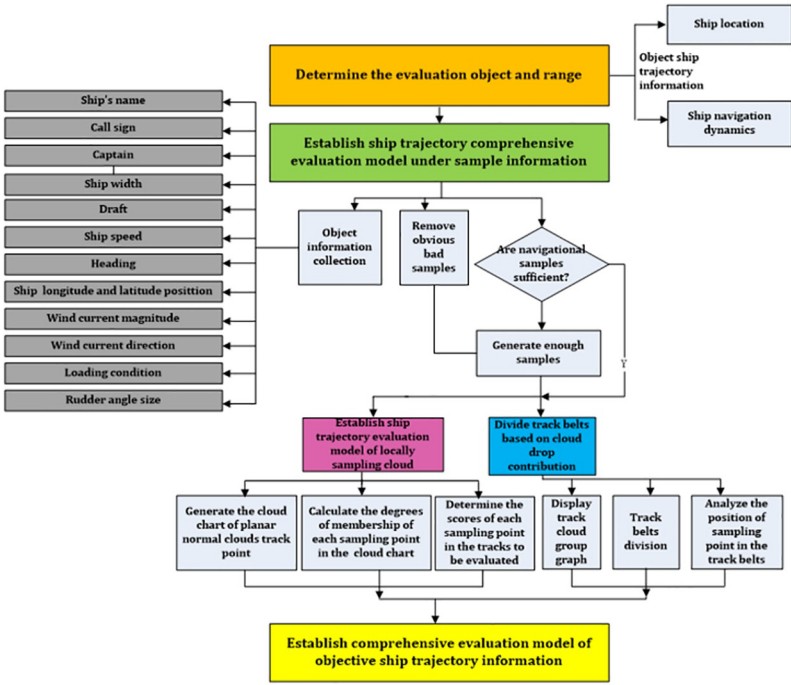

**Fig 3. The flow chart of ship trajectory evaluation based on sample information.**

1. Preparation: Determining the evaluation object and range. The evaluation object is the objective track information of the inward-port single ship left in the electronic chart, mainly including the dynamic information of ship position and navigation.

2. Sample information collection and analysis: The objective information involved in inward-port single-ship navigation training or examination in the electronic chart of the ship-handling simulator includes data such as the ship's name, call sign, captain, ship width, draft, ship speed, heading, ship longitude and latitude position, wind current magnitude, wind current direction, loading condition, and rudder angle size that are stored in a relational database. The obviously bad samples are removed according to the judgments of the assessors or the evaluation of experts. Furthermore, more sample information is generated by using a track point generation algorithm [38] if the sample quantity is insufficient, thus confirming the objective information of track samples.

3. Establish a local sampling cloud-track evaluation model: Setting longitude and latitude as axes, take local samples for each track line in equal time intervals to form sets by integrating track points at the same time in each track line. Then, the cloud chart model of this moment can be calculated using the backward normal cloud algorithm of uncertainty. Similarly, a cloud chart model of the planar forward normal cloud-track points of each moment can be generated. After local sampling of the unevaluated track in equal time intervals, calculate the membership of each sampling point in the planar forward normal cloud chart in the hundred-mark system, determine the planar normal cloud chart score of each sampling point of the unevaluated track, and finally form the track evaluation model of the planar normal forward cloud chart based on local sampling.

4. Because there may be a large difference between the cloud chart distribution generated by the track evaluation model and the actual channel, the track belts are divided by the cloud drop contribution based on the distribution feature of the normal cloud, and the widths of the track belts are computed and compared with the channel dimension.

5. Integrating the evaluation model of a planar normal cloud based on local sampling with the belt division method based on the cloud drop contribution of a normal cloud, and a comprehensive track evaluation model based on sample information is established.

## 3 Simulation and results

Taking M.V. DAQING 257 unloaded into Dalian Port as an example, the key cloud drop track belt, the basic cloud drop track belt, and the peripheral track belt can be divided according to the analysis in Section 2, as shown in Figs 4 and 5. The covering widths of the three track belts change with points at different locations during ship navigation. The track belts of the straight voyages are relatively centralized. If the track points are out of the channel under normal conditions of straight voyage (except emergency actions or other specific causes), the navigation is considered to be a failure to some extent. In this case, it is reasonable to adopt one vote veto or provide a failure treatment or take reduction measures conditionally. If the track points are divergent in a curved voyage (which may be out of the channel), it is considered as a normal case for the specificity of the curved channel owing to the influence of the cycle performance and vehicle and rudder performance of the ship. Because the widths of the curved and straight channels are the same in channel planning, it is possible for the ship to navigate out of the channel. Consequently, it is advisable to set a reduction scheme according to the actual demand and the requirement of assessors and integrate it into the track evaluation model of

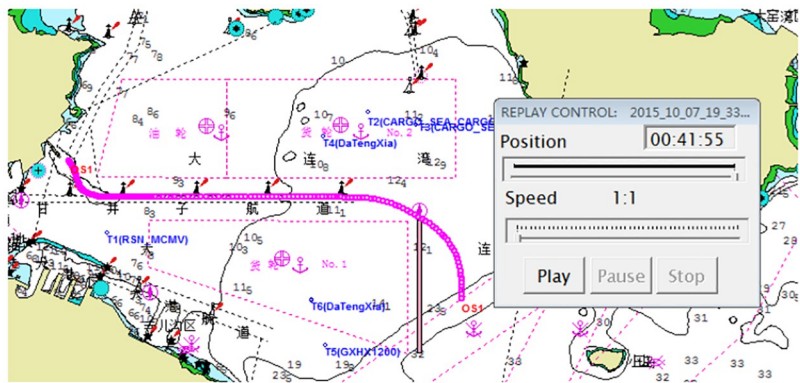

**Fig 4. $H_d$ in the electronic chart.**

an inward-port single ship based on a local sampling cloud chart. Thus, the intelligent track evaluation model based on sample information is proposed as follows:

Input: the unevaluated track $H_d$ and 20 objective samples track stored in the database.

Output: the unevaluated track score $H_{dscore}$.

**(1)** Objective track data sampling. Set north-east wind $1'$, east current as $0.5'$, and the initial navigation position as north latitude of $38°55.812'$ and east longitude of $121°47.028'$. Choose a navigation process of 60 min and take 127 track sampling points in the track line with equal time intervals (except the starting point). To avoid large errors, unreasonable track lines are removed. Therefore, Point $\beta$ of track line $\alpha$ can be represented as:

$$point_{\alpha\beta}, \alpha = 1, 2, \cdots, N, \beta = 1, 2, \cdots, M \tag{17}$$

**(2)** Generate a planar forward normal cloud chart of the track sampling point. For ease of calculation, the values of longitude and latitude are precise, all the longitude data are subtracted by $121°$, all the latitude data are subtracted by $38°$, and the minute levels are added $60'$ separately if there are ship positions of $122°$ or $39°$. For instance, define Point 14 as $point_\alpha(\varphi_{14}, \lambda_{14})$ using longitude and latitude of ship location, and combine all the 14th points of 20 objective samples track to form the following set.

$$S_{14} = \{point_1(\varphi_{14}, \lambda_{14}), point_2(\varphi_{14}, \lambda_{14}), \cdots, point_{20}(\varphi_{14}, \lambda_{14})\}$$

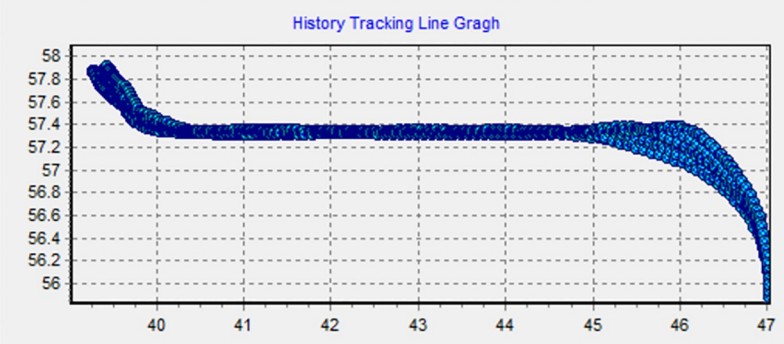

**Fig 5. History tracking line graph of this example.**

Assuming that $k = 1000$ track points are generated based on Set $S_{14}$, the following can be calculated:

$$E\hat{x}_{14} = \overline{\varphi}_{14} = \frac{1}{20}\sum_{\alpha=1}^{20}\varphi_{\alpha14} = 56.6075', E\hat{y}_{14} = \overline{\lambda}_{14} = \frac{1}{20}\sum_{\alpha=1}^{20}\lambda_{\alpha14} = 46.8265$$

$$En\hat{x}_{14} = \sqrt{\frac{\pi}{2}}\frac{1}{20}\sum_{\alpha=1}^{20}|\varphi_{\alpha14} - \overline{\varphi_{14}}| = 0.0122, En\hat{y}_{14} = \sqrt{\frac{\pi}{2}}\frac{1}{20}\sum_{\alpha=1}^{20}|\lambda_{\alpha14} - \overline{\lambda_{14}}| = 0.0212$$

$$(S\hat{x}_{14})^2 = \frac{1}{19}\sum_{\alpha=1}^{19}(\varphi_{\alpha14} - \overline{\varphi_{14}})^2, (S\hat{y}_{14})^2 = \frac{1}{19}\sum_{\alpha=1}^{19}(\lambda_{\alpha14} - \overline{\lambda_{14}})^2$$

$$He\hat{x}_{14} = \sqrt{(S\hat{x}_{14})^2 - (En\hat{x}_{14})^2} = 0.026, He\hat{y}_{\beta} = \sqrt{(S\hat{y}_{\beta})^2 - (En\hat{y}_{\beta})^2} = 0.0361$$

Thus, the forward normal planar cloud model of sampling point No.14 can be represented as follows:

$$\begin{aligned}Cloud_{14} &= (E\hat{x}_{14}, E\hat{y}_{14}, En\hat{x}_{14}, En\hat{y}_{14}, He\hat{x}_{14}, He\hat{y}_{14})\\ &= (56.6075, 46.8265, 0.0122, 0.0212, 0.026, 0.0361)\end{aligned}$$

Based on this model, a planar normal random number $(En\hat{x}'_{14}, En\hat{y}'_{14})$ is generated with the expected value of $(En\hat{x}_{14}, En\hat{y}_{14}) = (0.0122, 0.0212)$ and variance of $(He\hat{x}_{14}, He\hat{y}_{14}) = (0.026, 0.0361)$. Subsequently, a planar normal random number $draop(\varphi_{14i}, \lambda_{14i})$ can be generated with the expected value as, $(E\hat{x}_{14}, E\hat{y}_{14}) = (56.6075, 46.8265)$ and the variance as $(En\hat{x}'_{14}, En\hat{y}'_{14})$. The cloud drop certainty $\mu_{14i}$ can be calculated as follows:

$$\mu_{14i} = \exp\left(-\left(\frac{(\varphi_{14i} - E\hat{x}_{14})^2}{2(En\hat{x}'_{14})^2} + \frac{(\lambda_{14i} - E\hat{y}_{14})^2}{2(En\hat{y}'_{14})^2}\right)\right) \tag{18}$$

Each sampling point needs to output $i = 100$ cloud drops, and then the cloud chart of sampling point No.14 can be generated by the forward normal planar cloud generation algorithm, as shown in Fig 6. Similarly, 127 planar forward normal cloud group charts of all sampling points in the track line can be obtained, as shown in Fig 7.

(3) Divide track belts based on normal cloud drop contribution. Both of cloud drops belonging to longitude interval $[Ex_{\beta} - 0.67En_{\beta}, Ex_{\beta} + 0.67En_{\beta}] = [E\hat{x}_{14} - 0.67En\hat{x}_{14}, E\hat{x}_{14} + 0.67En\hat{x}_{14}]$ and latitude interval $[Ey_{\beta} - 0.67Eny_{\beta}, Ey_{\beta} + 0.67Eny_{\beta}] = [E\hat{y}_{14} - 0.67En\hat{y}_{14}, E\hat{y}_{14} + 0.67En\hat{y}_{14}]$ are in the key track belt. As in the field of navigation, $1'$ is approximately 1852 m, the widest distance in the latitude axis is $2 \times 0.67En\hat{y}_{14} \approx 0.0284'$, which is approximately 52.60 m. Similarly, it can be calculated that the widest distance in the longitudinal axis $2 \times 0.67En\hat{x}_{14} = 0.0163'$, that is 30.19 m.

Both the cloud drops belonging to longitude interval $[Ex_{\beta} - En_{\beta}, Ex_{\beta} + En_{\beta}] = [E\hat{x}_{14} - En\hat{x}_{14}, E\hat{x}_{14} + En\hat{x}_{14}]$ and latitude interval $[Ey_{\beta} - Eny_{\beta}, Ey_{\beta} + Eny_{\beta}] = [E\hat{y}_{14} - En\hat{y}_{14}, E\hat{y}_{14} + En\hat{y}_{14}]$ are in the basic track belt. The widest distance in the latitude axis is $2 \times En\hat{y}_{14} = 0.0424'$ approximately, that is 78.52m. In the same way, it can be calculated that the widest distance in the longitudinal axis is $2 \times En\hat{x}_{14} = 0.0244'$, that is 45.19 m.

Both the cloud drops belonging to longitude interval $[Ex_{\beta} - 2En_{\beta}, Ex_{\beta} + 2En_{\beta}] = [E\hat{x}_{14} - 2En\hat{x}_{14}, E\hat{x}_{14} + 2En\hat{x}_{14}]$ and latitude interval $[Ey_{\beta} - 2Eny_{\beta}, Ey_{\beta} + 2Eny_{\beta}] =$

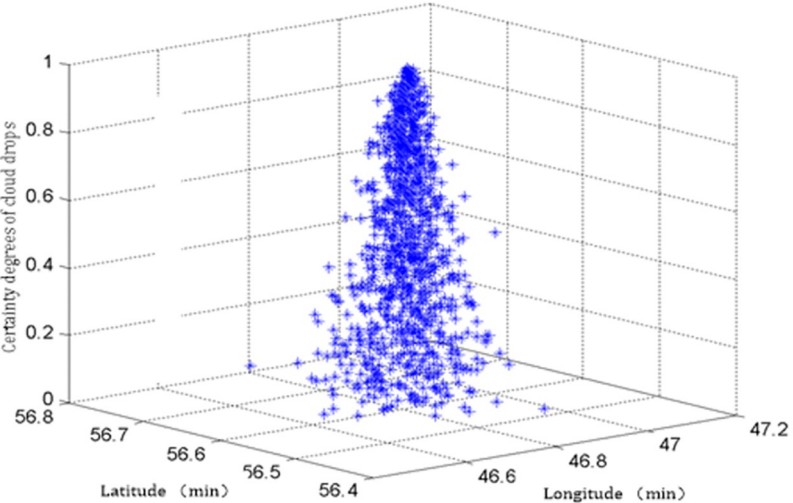

**Fig 6. The normal cloud diagram of sampling point No.14.**

$[E\hat{y}_{14} - 2En\hat{y}_{14}, E\hat{y}_{14} + 2En\hat{y}_{14}]$ are in the peripheral track belt. The widest distance in the latitude axis is $2 \times 2En\hat{y}_{14} = 2 \times 2 \times 0.0212 = 0.0848'$, that is 157.04 m. Similarly, it can be calculated that the widest distance in the longitude axis is $2 \times 2 \times En\hat{x}_{14} = 2 \times 2 \times 0.0122 = 0.0488'$, that is 90.38 m.

The channel width is approximately 0.15 nautical miles (277.8 m) in the electronic chart. Since the boundary of in-out-channel is set as the half position of the channel center line, the channel width for ship navigation is 138.9 m, while the width of the basic track belts in the latitude axes is 78.52 m, and the width of the peripheral track belts in the latitude axes is 157.04 m. The channel width is between the width of the basic track belts and the peripheral track belts in the latitude axes. If sample point No.14 of the unevaluated track belongs to the key track belt or the basic track belt, the score can be obtained directly by the cloud drop certainty. If sampling point No.14 of the unevaluated track belongs to the peripheral track belt, but not out of the channel central line or right line, the score can still be obtained directly by the cloud drop certainty. If sampling point No.14 of the unevaluated track belongs to the peripheral track belt, but out of the channel central line or right line, it is advisable to adopt the reduction measure. For strict requirements, assessors can specify that the maximum sampling point number out of the channel is 10, exceeding which the evaluation is disqualified. To relax the

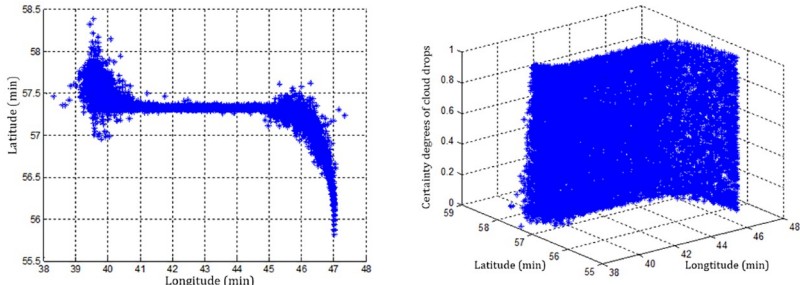

**Fig 7. The normal cloud group diagram of all sampling points in the trajectory.**

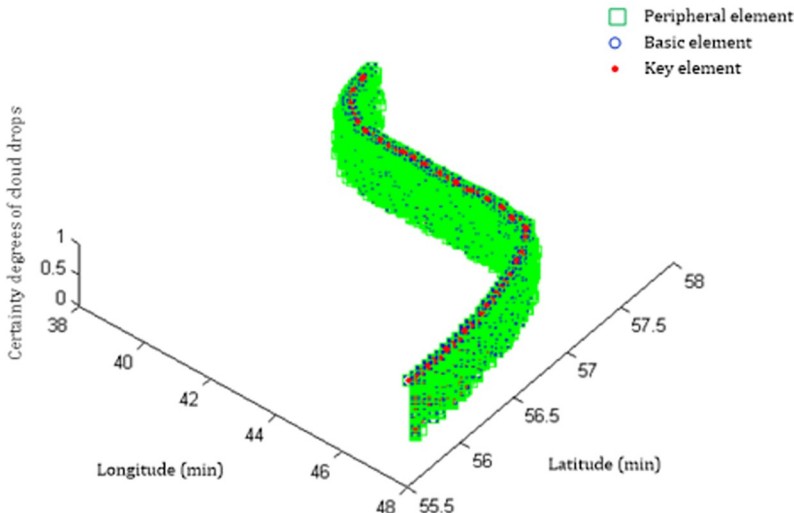

**Fig 8. The comparison chart of the three track belts.**

requirement, the weight of the evaluation score can be reduced proportionally or the evaluation score can be obtained directly (the vertical coordinate value is relatively smaller in the planar normal expected surface because it exceeds three main track belts, such as computing sampling point No.14 in Step (4)), while the evaluation is disqualified once the ship location is out of the width range of $2 \times 3 \times En\hat{x}_{14} \times 1852 = 235.57m$. The evaluation scheme can be established in detail based on actual demand. The distributions of the three track belts are shown in Fig 8.

**(4)** Obtain the evaluation results of the unevaluated track based on the sample information.

First, 127 track sampling points are taken in the unevaluated track line $H_d$ with equal time intervals. For easy calculation, the values of longitude and latitude are precise, all the longitude data are subtracted by 121˚, all the latitude data are subtracted by 38˚, and the minute levels are added 60′ separately if there are ship positions of 122˚ or 39˚. Then, sampling point No.14 is denoted as $point_D(\phi_{D14}, \lambda_{D14}) = point_D(56.573, 46.850)$.

Second, we analyze and determine the reference planar forward normal cloud surface. If the expected surface is chosen as the surface function of the planar forward normal cloud shown in Fig 9, the cloud drop certainty value of sampling point No.14 can be calculated as

$$
\begin{aligned}
Z_{D14} &= \exp\left(-\left(\frac{(\varphi_{D14} - Ex_{14})^2}{2(Enx_{14})^2} + \frac{(\lambda_{D14} - Ex_{14})^2}{2(Eny_{14})^2}\right)\right) \\
&= \exp\left(-\left(\frac{(56.573 - 56.6075)^2}{2(0.0122)^2} + \frac{(46.850 - 46.8265)^2}{2(0.0212)^2}\right)\right) \\
&= 0.00924
\end{aligned}
$$

For direct scoring, the score for sampling point No.14 is $Z_{D14} \times 100 = 0.9924$. Similarly, the score of the maximum boundary surface and the corrected expected surface as the reference standard are as shown in Figs 10 and 11, respectively. According to (3), the ship is located in the intermediate belt of the basic track belt and the peripheral track belt. Compared with the

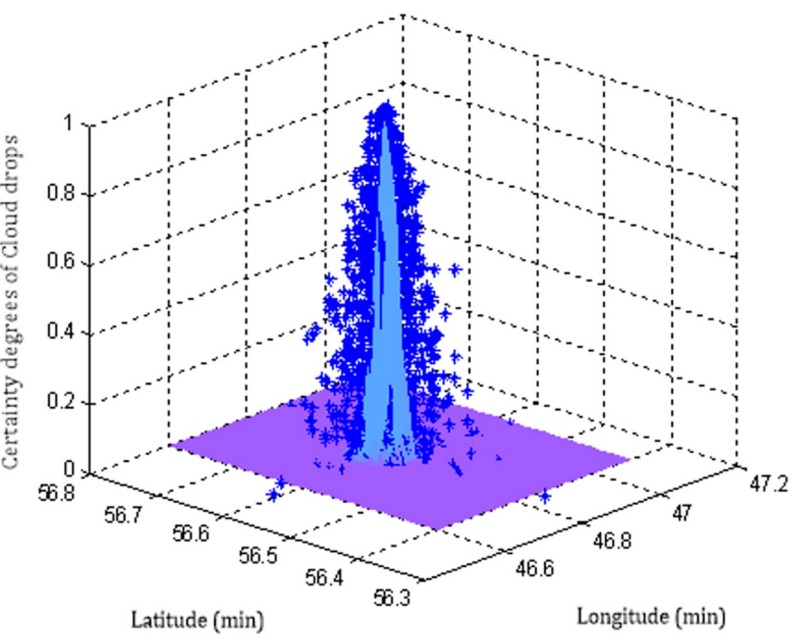

**Fig 9. The expected surface of sampling point No.14.**

three computation results, the result of the corrected expected surface is the most reasonable.

$$Zmax_{D14} = \exp\left(-\left(\frac{(\varphi_{D14} - Ex_{14})^2}{2(Enx_{14} + 3He\hat{x}_{14})^2} + \frac{(\lambda_{D14} - Ex_{14})^2}{2(Eny_{14} + 3He\hat{y}_{14})^2}\right)\right)$$

$$= \exp\left(-\left(\frac{(56.573 - 56.6075)^2}{2(0.0122 + 3 \times 0.026)^2} + \frac{(46.850 - 46.8265)^2}{2(0.0212 + 3 \times 0.0361)^2}\right)\right)$$

$$Z_{RD14} = \exp\left(-\left(\frac{(\varphi_{RD14} - Ex_{14})^2}{2(Enx_{14} + 3He\hat{x}_{14})^2} + \frac{(\lambda_{RD14} - Ex_{14})^2}{2(Eny_{14} + 3He\hat{y}_{14})^2}\right)\right)$$

$$= \exp\left(-\left(\frac{(56.573 - 56.6075)^2}{2(0.0122 + 0.026)^2} + \frac{(46.850 - 46.8265)^2}{2(0.0212 + 0.0361)^2}\right)\right)$$

Similarly, the normal corrected expected surfaces of sampling points No.8 to 16 can be generated subsequently, and the scores of each sampling point can be calculated by using the track evaluation model of an inward single ship based on a local cloud chart, as shown in Fig 12.

**(5)** Calculate the comprehensive evaluation results for the unevaluated track $H_{dscore}$. The track evaluation score of the 127 sampling points is $H_{dscore} = \frac{1}{127}\sum_{\beta=1}^{127} Z_{RD\beta} \times 100 = 76.28$. The scores of each sampling point and the total score of the unevaluated track are shown in Table 1 and Fig 13.

In this section, the evaluation result is obtained by using the intelligent track evaluation model based on local sampling through the sampling analysis of an unevaluated track. The evaluation result is consistent with that of experts based on subjective information, indicating the effectiveness of the proposed method. Furthermore, this method is completely computed

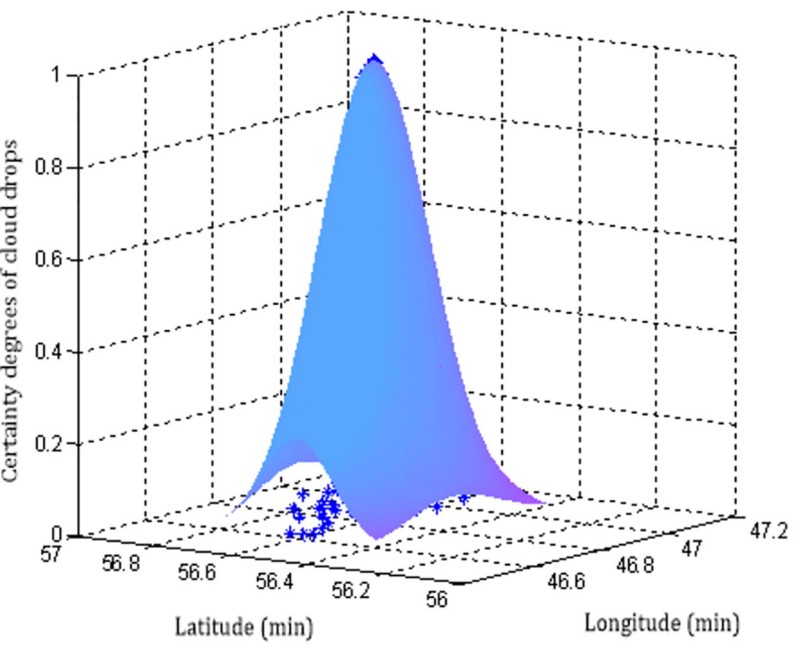

**Fig 10. The maximum boundary surface of sampling point No.14.**

from objective data; it is a certain advantage of objective evaluation application compared with the evaluation method based on subjective information.

## 4 Discussion

This study established a local sampling cloud-track evaluation model based on sample information to quantitatively evaluate the inward-port single-ship trajectory based on a ship-

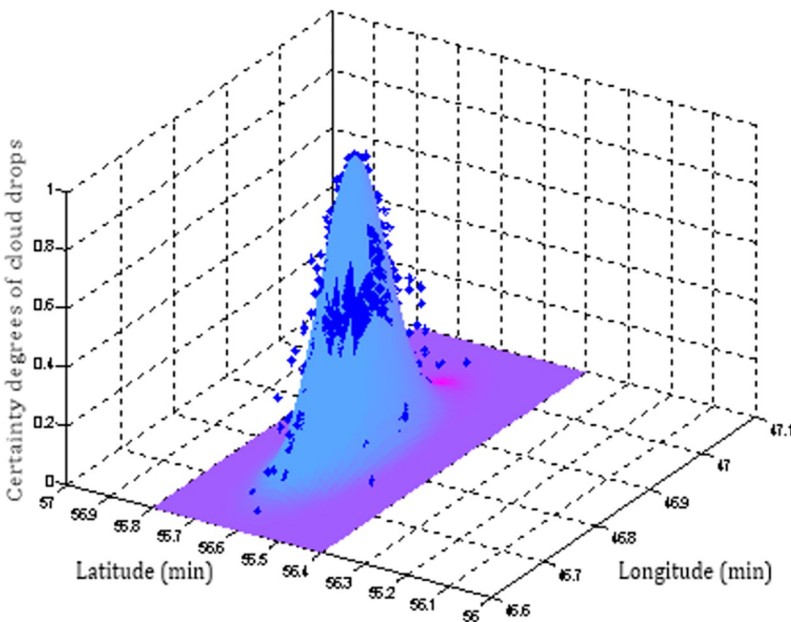

**Fig 11. Corrected expected surface of sampling point No.14.**

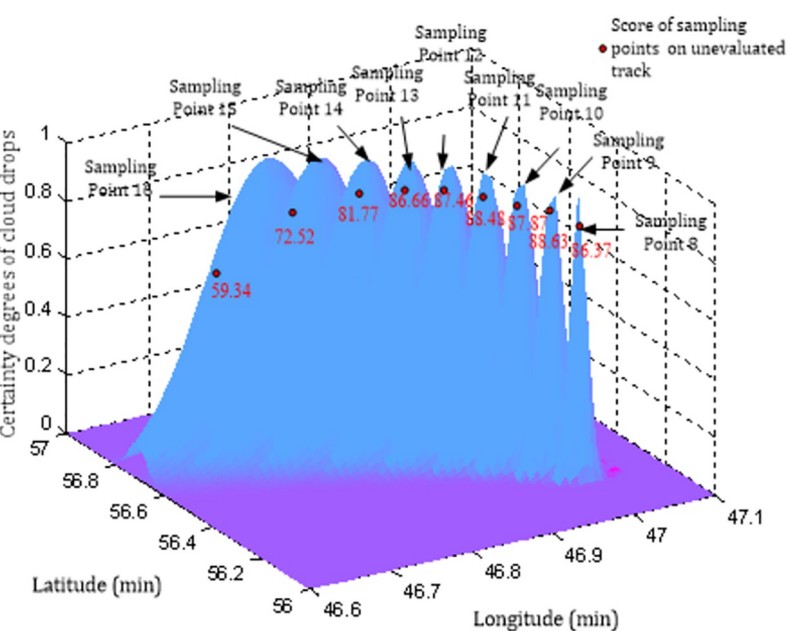

**Fig 12. Modified expected surfaces of sampling point No.8–16.**

handling simulator. First, we establish a track evaluation model for an inward-port single ship based on a local sampling cloud chart. Then, we analyze the track belt division based on the cloud drop contribution of the normal cloud. Finally, we establish comprehensive track evaluation model based on sample information. An example of M.V. DAQING 257 unloaded into Dalian Port is given to demonstrate the effectiveness of the model.

**Table 1. Scores of sampling points on the unevaluated track.**

| No. | Score | No. | Score | No. | Score | No. | Score | No. | Score | No. | Score | No. | Score |
|-----|-------|-----|-------|-----|-------|-----|-------|-----|-------|-----|-------|-----|-------|
| 1 | 100 | 19 | 63.91 | 37 | 95.24 | 55 | 58.39 | 73 | 62.31 | 91 | 92.32 | 109 | 84.13 |
| 2 | 98.97 | 20 | 61.09 | 38 | 96.81 | 56 | 46.68 | 74 | 63.35 | 92 | 92.71 | 110 | 83.95 |
| 3 | 93.88 | 21 | 58.43 | 39 | 96.65 | 57 | 54.34 | 75 | 65.83 | 93 | 92.93 | 111 | 83.99 |
| 4 | 84.23 | 22 | 56.19 | 40 | 96.35 | 58 | 55.18 | 76 | 65.66 | 94 | 93.62 | 112 | 83.81 |
| 5 | 96.26 | 23 | 54.58 | 41 | 91.58 | 59 | 56.23 | 77 | 65.97 | 95 | 97.54 | 113 | 83.22 |
| 6 | 92.97 | 24 | 49.54 | 42 | 88.21 | 60 | 55.67 | 78 | 67.93 | 96 | 96.86 | 114 | 82.75 |
| 7 | 90.15 | 25 | 55.14 | 43 | 95.78 | 61 | 56.19 | 79 | 67.13 | 97 | 94.1 | 115 | 81.63 |
| 9 | 88.63 | 27 | 67.25 | 45 | 95.53 | 63 | 56.84 | 81 | 68.94 | 99 | 88.84 | 117 | 79.04 |
| 10 | 87.87 | 28 | 70.76 | 46 | 92.32 | 64 | 53.19 | 82 | 70.38 | 100 | 87.72 | 118 | 76.78 |
| 11 | 88.48 | 29 | 72.6 | 47 | 91.16 | 65 | 53.08 | 83 | 70.35 | 101 | 85.62 | 119 | 74.04 |
| 12 | 87.46 | 30 | 74.5 | 48 | 89.24 | 66 | 57.11 | 84 | 75.44 | 102 | 84.45 | 120 | 70.89 |
| 13 | 86.66 | 31 | 74.29 | 49 | 84.14 | 67 | 56.03 | 85 | 78.57 | 103 | 82.58 | 121 | 67.45 |
| 14 | 81.77 | 32 | 78.85 | 50 | 73.49 | 68 | 63.13 | 86 | 80.7 | 104 | 78.41 | 122 | 63.61 |
| 15 | 72.52 | 33 | 83.17 | 51 | 75.65 | 69 | 64.27 | 87 | 85.27 | 105 | 82.18 | 123 | 68.67 |
| 16 | 59.34 | 34 | 86.84 | 52 | 75.78 | 70 | 65.48 | 88 | 87.76 | 106 | 83.31 | 124 | 46.58 |
| 17 | 65.54 | 35 | 90.5 | 53 | 69.87 | 71 | 64.43 | 89 | 88.95 | 107 | 83.83 | — | — |
| 18 | 66.4 | 36 | 93.67 | 54 | 63.89 | 72 | 64.04 | 90 | 89.94 | 108 | 84.27 | 127 | 57.93 |

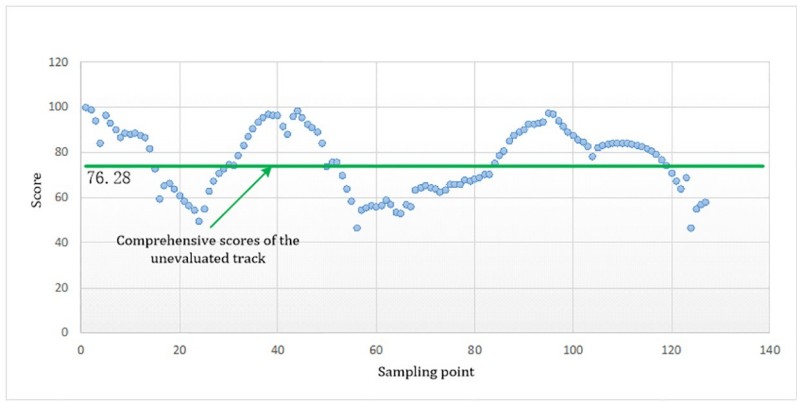

**Fig 13. Comprehensive scores of the unevaluated track.**

## 4.1 Theoretical and practical implications

First, in voyage training or examination of inward-port single ship using the ship-handling simulator, the track information is an important comprehensive embodiment of ship motion, involving objective navigation information of ship speed, heading, ship longitude and latitude position, wind current magnitude, wind current direction, loading condition, and rudder angle size [3]. Compared with subjective information, objective information is more stable, objective, and robust [34]. Therefore, to achieve more reasonable evaluation results, it is advisable to make full use of objective track information.

Second, based on objective information, cloud model theory considers the randomness, imperfection, and incompleteness of uncertain information [29], of which the backward normal cloud generator algorithm is one of the most important algorithms [32]. The backward cloud learning algorithm makes intelligent evaluation based on big data possible by inferring three key parameters of the cloud model from limited objective track sample information and generating large sample data by using a forward cloud generator.

Third, because ship positions and tracks are the obvious results of ship handling [37], it is an effective method to score ship navigation by comparing the position of the ship and that of the same sampling point in history sampling data. However, there may be a large difference between the cloud chart distribution generated by the track evaluation model and the actual channel. As a supplement, dividing the corresponding track belt precautionary area is another favorable approach for navigation evaluation and sailor reminder, where the track belts are divided by the cloud drop contribution based on the distribution feature of the normal cloud, and the widths of the track belts are computed and compared with the channel dimension.

## 4.2 Limitations and future research directions

Despite the above-mentioned advantages of the study, several limitations still exist. Further research can be undertaken on the following aspects. First, establishing a track index system is necessary to test an intelligent algorithm. In the present study, this was achieved by investing many captains and navigation experts. For the higher accuracy and reasonability considerations of the evaluation results, more people should be investigated. Second, for intelligent evaluation using machine learning based on massive data, the data are not available at the present stage. In this study, mass data were generated using the track point generation algorithm, and the corresponding parameters were calculated using the backward cloud algorithm. For

further study, establishing an internet database of the ship-handling simulator evaluation could be more desirable through cooperation with national maritime colleges, marine boards, and crew training organizations. In addition, the influence of ship speed on the sampling points should also be considered.

## 5 Conclusions

In this study, a comprehensive track evaluation model of a local sampling cloud based on sampling information is established for the navigation process of an inward-port single ship in a ship-handling simulator. The summary of this study is as follows:

1. To evaluate the practical ability of crew more reasonably, a track evaluation model of a planar forward normal cloud chart under sample information based on the forward normal cloud generator and the backward normal cloud generator is proposed.

2. Focused on the problem of divergence of track sampling cloud, a track belt division method based on the contributions of normal cloud drops is proposed.

3. Combining the above track evaluation model with the track belt division method, a comprehensive track evaluation scheme of the local sampling cloud based on sampling information is established.

4. An example of M.V. DAQING 257 unloaded into Dalian Port is given to demonstrate the effectiveness of the model, with the result that it is consistent with expert evaluation results based on subjective information. Thus, the proposed model provides a new approach for intelligent evaluation under sample information.

## Supporting information

**S1 Data.**
(XLSX)

## Acknowledgments

We appreciate the data support from the Fujian Maritime Safety Administration (MSA), China.

## Author Contributions

**Conceptualization:** Cheng Fang, Hongxiang Ren, Jianjun Wu.

**Data curation:** Jianjun Wu.

**Methodology:** Cheng Fang, Hongxiang Ren.

**Software:** Cheng Fang.

**Supervision:** Xinguo Liu.

**Validation:** Cheng Fang, Xinguo Liu.

**Writing – original draft:** Wei Zhou.

**Writing – review & editing:** Cheng Fang, Jianjun Wu.

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
