## [Decision Letter · Decision Letter 0]

17 Nov 2021

PONE-D-21-23286Towards an uncertainty evaluation model for marine track considering local sampling cloud and sample informationPLOS ONE

Dear Dr. Wu,

Thank you for submitting your manuscript to PLOS ONE. After careful consideration, we feel that it has merit but does not fully meet PLOS ONE’s publication criteria as it currently stands. Therefore, we invite you to submit a revised version of the manuscript that addresses the points raised during the review process.

We look forward to receiving your revised manuscript.

Kind regards,

Yiming Tang, Ph.D.

Academic Editor

PLOS ONE

“This research was supported by the Zhejiang Provincial Natural Science Foundation of China under Grant No. LGG18E090001, the Zhejiang Provincial Postdoctoral Research Project under Grant 225846, and the Open Research Project of the State Key Laboratory of Industrial Control Technology, Zhejiang University, China (No. ICT2021B16).”

“This research was supported by the Zhejiang Provincial Natural Science Foundation of China under Grant No. LGG18E090001, the Zhejiang Provincial Postdoctoral Research Project under Grant 225846, and the Open Re-search Project of the State Key Laboratory of Industrial Control Technology, Zhejiang University, China (No. ICT2021B16).”

4. Please remove your figures from within your manuscript file, leaving only the individual TIFF/EPS image files, uploaded separately.  These will be automatically included in the reviewers’ PDF.

Reviewers' comments:

Reviewer's Responses to Questions

**Comments to the Author**

1. Is the manuscript technically sound, and do the data support the conclusions?

Reviewer #1: Yes

Reviewer #2: Yes

2. Has the statistical analysis been performed appropriately and rigorously? 

Reviewer #1: Yes

Reviewer #2: Yes

3. Have the authors made all data underlying the findings in their manuscript fully available?

Reviewer #1: Yes

Reviewer #2: No

4. Is the manuscript presented in an intelligible fashion and written in standard English?

Reviewer #1: Yes

Reviewer #2: Yes

5. Review Comments to the Author

Reviewer #1: The research paper was written well. authors explained the cloud techniques in good and understand way. really review this manuscript need lot patients . i appreciate the authors for used all formula.

Reviewer #2: Feng et al. submitted the manuscript entitled “Towards an uncertainty evaluation model for marine track considering local sampling cloud and sample information” in PLOS ONE journal. In this study, the authors established a local sampling cloud-track evaluation model based on sample information to quantitatively evaluate the inward-port single-ship trajectory based on a ship-handling simulator. To achieve the objectives, the authors at first establish a track evaluation model for an inward-port single ship based on a local sampling cloud chart. Subsequently, the authors analyzed the track belt division based on the cloud drop contribution of the normal cloud. Finally, the authors establish comprehensive track evaluation model based on sample information.

The manuscripts outcomes would helpful for the navigation field. I would recommend the revision before the final acceptance. Some of my comments are listed as follows-

1. There is an area of improvement of the discussion section. The arguments in the discussion section is not supported by the any citation of the previous works.

2. Less people are investigated, which may affect the precision of the evaluation result

6. PLOS authors have the option to publish the peer review history of their article (what does this mean?). If published, this will include your full peer review and any attached files.

Reviewer #1: No

Reviewer #2: No

---

## [Author Response · Author response to Decision Letter 0]

6 Dec 2021

We are very grateful to Dr. Yiming Tang, the Academic Editor, the Reviewers, and the magazine staff for their constructive comments and suggestions to improve the quality of our manuscript.

The manuscript has been changed to the format of Latex as required.

The manuscript has been revised according to the comments, and the corresponding changes can be found in the file of ‘Revised Manuscript with Track Changes’. Detailed modifications of the manuscript and answers to the questions are listed in the file of 'Response to Reviewers'.

---

## [Decision Letter · Decision Letter 1]

17 May 2022

Towards an uncertainty evaluation model for marine track considering local sampling cloud and sample information

PONE-D-21-23286R1

Dear Dr. Wu,

We’re pleased to inform you that your manuscript has been judged scientifically suitable for publication and will be formally accepted for publication once it meets all outstanding technical requirements.

Kind regards,

Yiming Tang, Ph.D.

Academic Editor

PLOS ONE

Additional Editor Comments (optional):

Reviewers' comments:

Reviewer's Responses to Questions

**Comments to the Author**

1. If the authors have adequately addressed your comments raised in a previous round of review and you feel that this manuscript is now acceptable for publication, you may indicate that here to bypass the “Comments to the Author” section, enter your conflict of interest statement in the “Confidential to Editor” section, and submit your "Accept" recommendation.

Reviewer #2: All comments have been addressed

2. Is the manuscript technically sound, and do the data support the conclusions?

Reviewer #2: Yes

3. Has the statistical analysis been performed appropriately and rigorously? 

Reviewer #2: Yes

4. Have the authors made all data underlying the findings in their manuscript fully available?

Reviewer #2: Yes

5. Is the manuscript presented in an intelligible fashion and written in standard English?

Reviewer #2: Yes

6. Review Comments to the Author

Reviewer #2: Authors have addressed all the comments very carefully. Therefore, I recommend the paper to publish in Plos ONe journal

7. PLOS authors have the option to publish the peer review history of their article (what does this mean?). If published, this will include your full peer review and any attached files.

Reviewer #2: No

---

## [Editor Report · Acceptance letter]

14 Jun 2022

PONE-D-21-23286R1 

Towards an uncertainty evaluation model for marine track considering local sampling cloud and sample information 

Dear Dr. Wu:

I'm pleased to inform you that your manuscript has been deemed suitable for publication in PLOS ONE. Congratulations! Your manuscript is now with our production department. 

Kind regards, 

on behalf of

Professor Yiming Tang 

Academic Editor

PLOS ONE